# Trends and Determinants of Operative Vaginal Delivery at Two Academic Hospitals in Johannesburg, South Africa 2005–2019

**DOI:** 10.3390/ijerph192316182

**Published:** 2022-12-03

**Authors:** Afikile Dutywa, Gbenga Olorunfemi, Langanani Mbodi

**Affiliations:** 1Department of Obstetrics and Gynaecology, Faculty of Health Science, School of Clinical Medicine, University of the Witwatersrand, Johannesburg 2000, South Africa; 2Division of Epidemiology and Biostatistics, School of Public Health, University of the Witwatersrand, Johannesburg 2000, South Africa

**Keywords:** operative vaginal delivery rates, instrumental delivery, caesarean section rates, vacuum, forceps, determinants of operative vaginal delivery, trends, join point, South Africa

## Abstract

Operative Vaginal delivery (OVD) can reduce perinatal and maternal morbidity and mortality especially in low resource setting such as South Africa. We evaluated the trends and determinants of OVD rates using join point regression at Charlotte Maxeke Johannesburg (CMJAH) and Chris Hani Baragwaneth (CHBAH) Academic Hospitals from 1 January 2005–31 December 2019 and conducted a comparative study of OVD (n = 179) and normal delivery (n = 179). Over the 15-year study period (2005–2019), 323,617 deliveries and 4391 OVDs were conducted at CHBAH giving an OVD rate of 1.36 per 100 births. In CMJAH, 74,485 deliveries and 1191 OVDs were conducted over an eleven-year period (2009–2019) with OVD rate of 1.60 per 100 births. OVD rate at CHBAH increased from 2005–2014 at 9.1% per annum and declined by 13.6% from 2014–2019, while OVD rates fluctuates at CMJAH. Of the 179 patients who had OVD, majority (n = 166,92.74%) had vacuum. Women who had OVDs were younger than those who vaginal delivery (*p*-value < 0.001). The prevalence of OVDs was higher among nulliparous women (*p*-value < 0.001), HIV negative women (*p*-value = 0.021), underweight (*p*-value < 0.001) as compared to normal delivery. The OVD rates has dramatically reduced over the study period This study heightens the need to further evaluate barriers to OVD use in our environment

## 1. Introduction

Operative Vaginal Delivery (OVD) is the application of either forceps or vacuum in order to shorten the second stage of labour and expedite the delivery to prevent maternal and perinatal morbidity and mortality [1,2,3,4,5]. The global incidence of OVDs varies between 1–15% [6]. Although, there is a global decline in OVD rates, however, obstetricians and other accoucheurs in high income countries (HICs) perform more OVD procedures (at a rate of 10–15%) as compared to the OVD rates performed by accoucheurs in low middle income countries (LMICs at 1–1.5%) [6,7,8,9,10,11,12]. In South Africa, the OVD rates appears to be low at 1–3% [8,13].

Many studies attributed declining trends of OVD rates in low and middle income countries (LMICs) to lack of appropriate skills and equipment, fear of litigation, poor supervision of junior or inexperienced birth attendants, and lack of relevant local guidelines and policies to entrench OVDs [1,3,6,7,8,14,15]. In South Africa, another contributory reason to downward trend to OVD rates was linked to previous guidelines of the prevention of mother to child transmission of HIV(PMTCT) program that discouraged use of vacuum among HIV positive patients [3,7]. There is also a misconception that instrumental delivery may be associated with increased risk of brain injury to the baby [3,7,16]. The most common instrument for OVD procedures in both LMICs and HICs is vacuum (as compared to forceps [6,15,17,18]. Thus, the vacuum extraction rate in LMICs is 3.9%, while forceps rate is 0.2% [6].

The World Health Organization (WHO) recommends that the optimal caesarian section rate should be between (10–15%) [19]. However, as OVD rates decreases, caesarean section rates have been increasing with associated increase in morbidity and mortality. In South Africa, the institutional caesarean section rates were between 27.2% and 50.6% [6,7,8,9,16,20,21]. A second stage caesarean section is more difficult to perform with significant maternal morbidity and mortality related to hemorrhage, extended hospital stay, bladder injury, and unintended extensions of the uterine incisions [1,3,5,16,20,22,23]. Generally, if OVD is performed by a skilled personnel, based on safety guidelines, the procedures can potentially reduce the incidence of maternal and foetal morbidity and mortality that are associated with short and long-term complications of performing a second stage caesarean section. Furthermore, OVD potentially reduces the health costs associated with caesarean section [6,7].

South Africa is an upper middle-income country and the health care system have undergone multiple improvement in service deliveries (especially access to maternity care) after the commencement of a multi-racial democracy in 1994 [24]. Furthermore, South Africa has one of the highest global prevalence of Human immunodeficiency virus infection (HIV) [25,26,27]. The aforementioned may impact on the trends in maternity and intrapartum care in the country [3]. Additionally, there is a notion that junior doctors prefer performing caesarean section in place of OVD because of lack of requisite skills [6,7,8,14,15]. The evidence of the current trends in OVD rates with its associated factors at the largest maternity centers in South Africa (Charlotte Maxeke Johannesburg Academic Hospital (CMJAH) and Chris Hani Baragwaneth Academic Hospital (CHBAH)) is not available. Hence, we aimed to evaluate the trends in the OVD rates and its associated factors in the two academic hospital in Johannesburg over a 15-year period in order to provide evidence for planning, training, improvement in maternity care and further contribute to global statistics of OVD trends.

## 2. Methodology

The study was a retrospective trends analysis and comparative cross-sectional study of patients who had OVD at CMJAH and CHBAH, Johannesburg, Gauteng Province, South Africa. These hospitals are quaternary hospitals. 

The annual total birth, live birth, caesarean section cases, OVDs, (stratified by OVD types (Vacuum, Forceps) were extracted from the labor ward and maternity records at CHBAH (from 1 January 2005 to 31 December 2019) and CMJAH (from 1 January 2005 to 31 December 2019) for analysis. 

Afterwards, consecutive 179 clinical files of women that had operative vaginal deliveries and consecutive 179 case files of women that had normal vaginal deliveries were retrieved after identifying them from the labor ward register. Information about socio-demographic characteristic (age, ethnicity, occupation), Obstetric factors (gravidity, parity, weight, height, booking bloods group, retroviral status), booking status and gestational age at booking and number of antenatal visits (registered, and medical conditions were extracted. Indications, types of OVD and intrapartum factors such as induction or spontaneous labor, duration of active labor (marked from 4 cm dilated), and cadre of the accoucheurs. 

### 2.1. Ethical Considerations 

Ethical approval for this study was obtained from the Human Research and Ethics Committee (HREC) of the University of the Witwatersrand before the commencement of the study (Ref number: M201031). Anonymity and confidentiality of the data was maintained. Approval was obtained from the Chief Executive officer of CMJAH and CHBAH before the commencement of the study.

### 2.2. Statistical Analysis

Data was entered on excel. The annual OVD and caesarean section rates were calculated by dividing the annual numbers by the total birth. The trends in OVD and caesarean section rates were then analyzed using the Join point regression software version 4.3.1 (National Cancer Institute). Poisson regression approach with a maximum of 4 join points and 4499 Monte Carlo permutation tests was conducted for the trend. The average annual percent change (AAPC) and annual percentage change (AAPC) of segmental trends was then obtained. If the APC is negative or positive, the trends were described as decreased or increased trends while values between −0.5 to +0.5 are described as stable (if the *p*-value > 0.05). The data for the comparative study was imported into Stata version 16 (Statacorp, College Station, TX, USA) statistical software for analysis. Categorical and continuous variables were respectively described using frequency and percentages, mean and standard deviation (or median and interquartile range if not normally distributed). Pearson’s chi square was used to assess the association between categorical variables and mode of delivery (OVD/normal delivery) while Student’s *t*-test or Mann Whitney U test was used to check the association between continuous variables and mode of delivery. Comparison between sociodemographic and clinical variables among the two hospital was also conducted. Univariable and multivariable binary logistic regression was conducted using backward elimination technique to further assess the socio-demographic and clinical characteristics and mode of delivery. Statistically significant level was set at *p*-value < 0.05. Two-tailed test of hypothesis was assumed. 

## 3. Results

### 3.1. Trends in Total Births, Caesarean Section and Operative Vaginal Deliveries in CHBAH and CMJAH 

Over the 15 years’ study period (2005–2019), 323,617 deliveries were conducted at CHBAH giving an average of 21,574 deliveries per annum. The hospital also performed 114,693 caesarean sections and 4391 OVDs with an average of 7646 caesarean sections and 292 OVDs per annum. The overall caesarean section and OVD rates at CHBAH were 35.44 and 1.36 per 100 births respectively. In CMJAH, 74,485 deliveries were conducted over an eleven-year period (2009–2019) giving an average of 6771 deliveries per annum. The hospital performed 37,326 caesarean sections and 1191 OVDs over eleven years at an average of 3397 caesarean sections and 108 OVDs per annum. Thus, the overall caesarean section and OVD rates were 50.17 and 1.60 per 100 births respectively (Appendix A).

### 3.2. Caesarean Section Trends

The caesarean section rate increased from 2005 to 2019 at 3.5% per annum (AAPC: 3.5%, 95%CI: 3.0% to 4.0%) in CHBAH (Figure 1A and Figure 2A, Table 1, Appendix A). Although the caesarean section rate was generally higher at CMJAH from 2009–2019, the overall trends was not a significant increase (4.1% per annum [AAPC: 4.1, 95%CI: −10.7 to 2.9, *p*-value = 0.2]). Joinpoint regression identified 3 trends of caesarean section rates at CMJAH as shown on Figure 1A and Figure 3A.

### 3.3. Operative Vaginal Delivery Trends

The overall trends of OVD at the two hospitals showed a statistically non-significant decline. The OVD rates at the two centres varied as reported in Figure 1B and Figure 3B, Table 1, Appendix A.

### 3.4. Forceps Delivery Trends

The rates of forceps delivery was generally about one-quarter of the rate of vacuum delivery in the two hospitals. Furthermore, the forceps delivery rates was generally higher at CMJAH as compared to the rates at CHBAH from 2015–2019. The forceps delivery rate at CHBAH declined from 0.35 per 100 births in 2005 to 0.09 per 100 births in 2009 at 27.8% per annum and a non-significant rise in rates subsequently occured from 2009–2012 with a final non-signficant decline of 19.5% per annum from 2012(1.39 per 100 births) to 2019 (0.92 per 100 births). (Figure 4 and Figure 5A, Table 1, Appendix A). However, in CMJAH, there was an initial non-significant rise in forceps delivery rate at 77.7% per annum from 0.25 100 births in 2009 to 0.89 per 100 births in 2011 and a subsequent decline of 14.4% per annum from 2011 to 0.19 per 100 births in 2019 (Figure 4 and Figure 5B, Table 1, Appendix A).

### 3.5. Vacuum Delivery Rate

The Vacuum delivery rates was generally higher at CMJAH as comapared to CHBAH from 2009–2014, and the rates became slightly higher at CHBAH from 2015–2019.

At CHBAH, the vacuum delivery rose at a rate of 9.3% per annum from 0.51 per 100 births in 2005 to 1.66 per 100 births in 2013 and there was a subsequent decline of about 9.4% per annum from 2013 to 0.93 per 100 births in 2019 (which approaches the rates of 2005). (Figure 4 and Figure 5B, Table 1, Appendix A). Similarly, in CMJAH, the vacuum delivery rates initially increased at non-significant rate of 10.4% per annum from 1.20 per 100 births in 2009 to 2.08 per 100 births and subsequently declined at non-significant rate of 13.8% per annum to 0.88 per 100 births in 2019. (Figure 4 and Figure 6B, Table 1, Appendix A). The trends in forceps and vacuum delivery rates at CMJAH was similar showing an initial rise and a subsequent decline. 

### 3.6. Relationship between Socio Demographic and Clinical Characteristics and Operative Vaginal Delivery 

The relationship between the socio-demographic characteristics and operative vaginal delivery were as described on Table 2. 

### 3.7. Association between Sociodemographic and Clinical Characteristics and Operative Vaginal Deliveries

There was an association between socio-demographic factors and OVD. OVD was commonly performed on younger women and the prevalence was higher in teenagers. No midwife conducted any OVD and the majority were conducted by registrars. Table 2 further describes these characteristics.

### 3.8. Determinants of Operative Vaginal Delivery

Table 3 describes the determinants of OVD after univariable and multivariable logistic regression. After univariable regression modelling, the risk factors for OVD deduced from the study were:–teenagers, nulliparity, prolonged second stage, induction of labor, right occipito-anterior position, normal or underweight. After multivariate analysis, employment and HIV status that were initially deduced as risk factors were no longer considered as risk factors. At univariable analysis, the odds of having OVD as compared to vaginal delivery among students was about 3-fold as compared to the odds of OVD among women who were employed (COR: 2.54, 95%CI: 1.20–5.37, *p*-value = 0.015). The odds of OVD among HIV negative women was about 1.9-fold as compared to the odds of OVD among HIV positive women. (COR: 1.89, 95%CI: 1.85–1.93, *p*-value < 0.001). Furthermore, the odds of OVD decreases with increasing age and there was no statistically significant difference in the risk of OVD among teenagers and women aged 20–24 years. (*p*-value = 0.491).

After multivariable regression modelling, the odds of OVD among women with parity between 1–3 was about 58% lesser as compared to the odds of OVD among nulliparous women. Similarly, the odds of OVD among women with 4 or more parities was about 87% lesser as compared to nulliparous women. BMI was also a predictor of OVD as the odds of OVD decreases with increasing BMI. However, after multivariable regression analysis, there was no difference between women with normal BMI and those who are considered overweight. 

In the second multivariable model, for every hour increase in duration of labor, the odds of having an OVD increased by 11% (AR OR: 1.11, 95%CI: 1.03–1.20, *p*-value = 0.004).

### 3.9. Comparison between Forceps and Vacuum Delivery 

#### Types of Operative Vaginal Delivery

Of the 179 patients who had operative vaginal delivery, majority (*n* = 166, 92.74%) had Vacuum (Kiwi Cup) delivery while a few (*n* = 11, 6.15%) had forceps delivery and only2 patients (1.12%) had sequential vacuum and forceps delivery. (Figure 7). However, all the 2 sequential vacuum and forceps deliveries were performed at CMJAH while the only Kielland’s delivery was performed at CHBAH (Appendix A).

Table 4 showed the comparison of the sociodemographic and clinical characteristics of women who had forceps or vacuum delivery only after excluding the two women who had sequential vacuum and forceps delivery.

There was an association between the cadre of the accoucheur and the type of OVD performed. Of the 11 forceps delivery that were performed, 27.3% were performed by the consultants while of the 166 Vacuum deliveries, only 3.61% were performed by the consultants. In contrast, the registrars performed 72.7% and 86.8% of forceps and vacuum deliveries respectively. Other variables were not statistically significant. However, it appears that more forceps deliveries were performed at CHBAH as compared to CMJAH while 18.2% of forceps deliveries were performed for preterm births while 3.01% of vacuum deliveries were performed for preterm births.

After multivariable regression, women 35 years and older had about 86% lesser odds of having vacuum delivery as compared to women who were younger than 35 years. After correcting for confounding variables, women with parity of 1 and above had about 7.5 times higher likelihood of having vacuum delivery as compared to forceps delivery. The odds of utilizing vacuum delivery for OVD among registrars and medical officers was about 7.1 times the odds of vacuum delivery use among the consultants (Table 5).

## 4. Discussion

This study aimed to evaluate the trends in the OVD rates over a 15-year period and its associated factors at two academic hospitals in Johannesburg with high volume deliveries in excess of 20,000 and 6000 deliveries per annum. To our knowledge, our study was the first to utilize Join point regression modelling to evaluate the trends of OVDs in sub–Saharan Africa over such a long period. Our study showed that despite having about thrice the volume of deliveries at CHBAH, the average annual OVD rate at CBHAH was lower than the rates at CMJAH (1.36 per 100 births vs. 1.60 per 100 births). Our reported tertiary hospital rates were generally comparable to the reported rates at some tertiary hospitals in South Africa (1–3%) and other LMICS (1–1.15%) [10,21,28,29]. However, the OVD rates from our study were higher than the reports from some hospitals in Kwazulu Natal province of South Africa (<1%) and Nigeria (0.4%) [3,12]. In contrast, the OVD rates of tertiary hospitals in HICs was about 10 times the rates at our study centers [6,7,8,9]. Globally, OVD trends appears to decline with simultaneous rise in caesarean section rates [10,21,28,29]. However, our study showed that there were increased rates of both caesarean section and OVD rates from 2005–2014 at CHBAH. But OVD rates declined in the last 6 years of the study (2014–2019) as the caesarean section rates continued to rise. Thus, other factors and indications such as prevention of mother to child transmission of HIV (PMTCT) and other hospital protocols might have been responsible for the initial rise in caesarean section rates despite a rise in OVD rates. The observed decline in OVD rate might have substantially contributed to the increasing caesarean section rates in the last 6 years at the center. The dramatic decline in OVD rates at CHBAH from 2014–2019 may be related to a number of factors. As previously reported, lack of requisite training, skills and supervision to perform OVD, lack of necessary equipment, and fear of litigation are some factors responsible for the declining art of OVD [3,6,7,8,13,30,31]. In South Africa, the initial guideline for the prevention of Mother to Child transmission of HIV (PMTCT) recommended that OVD, (especially Vacuum delivery) should be avoided among HIV positive women [3]. Such guidelines might have contributed substantially to reduced OVD rates since South Africa has high prevalence of HIV at 57.8% and 2017 it was estimated that about 7.7 million of population is living with HIV [26]. Indeed, our cross-sectional study revealed that the likelihood of performing OVDs among HIV negative women was about twice the odds of performing OVDs among HIV positive women. This study therefore highlights the need for the hospital to institute deliberate training processes to further investigate the barriers to OVD in properly selected cases. Furthermore, opportunity for training in the conduct of OVD should be enhanced at the hospital [3,6,7,10,12,28,32].

Although caesarean section and OVD rates at CMJAH were higher than at CHBAH, their trends generally showed an initial increase, a subsequent decline and a latter rise in the last four years of the study (2016–2019). Thus, the generally reported negative correlation between institutional caesarean section rates and OVD rates is not very apparent at CMJAH. The latter increase in OVD trends at CMJAH may suggest that the institution is reviewing their protocols to ensure that resident doctors are trained and thereby develop confidence in the conduct of OVDs. Such deliberate protocol should be further sustained and reviewed periodically. Moreover, the current PMTCT guidelines and the free roll-out of Antiretroviral treatment in the country coupled with clear guidelines encouraging strategies to increased training of birth attendants on use of OVD in the country might also assist to sustain the upward trends of OVDs at the hospital [3,26,33,34,35]. Studies have also shown a decline in OVD rates in the HIC. However, such decline is not as sharp as in LMIC such as our studied hospitals.

The caesarean section rates at CHBAH (26.5–45.52%) and CMJAH (42.70–72.75%) is relatively high as compared to the recommendation of the world health organization (10–15%) [36]. Global incidence of caesarean section is 21.1% which is higher than WHO recommendation [19,30] These quaternary hospital rates are comparable to a reported tertiary hospital rates from Johannesburg (50.6%), Egypt (67.3%), Eastern Asia, Western Asia, and Northern Africa (44.9%, 34.7%, 31.5% point increase, respectively) [21,36,37]. The caesarean section rates were relatively high at our centers as is the case in other tertiary institution in South Africa [38]. These hospitals by virtue of being referral centers, largely managed high risk pregnancies and acute emergencies that were transferred from several other lower-tiered hospitals in the Gauteng province and other contiguous provinces of South Africa. Some studies suggest that the decline in OVDs can partly be explained by the increasing trends in caesarean section [10,21,39]. Furthermore, some hospitals and practitioners may prefer to perform caesarean section instead of OVD because of extra revenue although our study was performed at the two tertiary hospitals that are publicly owned and patients do not pay for services.

From the trend analysis the rates of forceps delivery was generally about one-quarter of the rate of vacuum delivery in the two hospitals. Further more than nine out of ten patients that had OVD in our cross-sectional study had vacuum (kiwi cup). Our finding is in keeping with other studies in LMICs that showed that birth attendants are likely to choose vacuum because of ease of use, and lack of expertise and supervision to perform forceps delivery [3,6,7,12,31,40]. Although Cochrane studies has shown that if forceps is used appropriately, it is likely to be successful. However it is associated with higher maternal morbidity if incorrectly used [41]. Our study revealed that there was an association between the cadre of the accoucheur and type of OVD that was performed. Consultant obstetricians tends to perform more forceps deliveries than vacuum delivery while the odds of performing vacuum delivery among registrars was about 7 folds as compared to the consultants. This findings showed lack of supervision and training of registrars in the use of forceps deliveries [9,12,31]. It is therefore pertinent to train the registrars in the use of forceps deliveries as majority of OVDs were performed by registrars. If registrars are well-trained in forceps delivery, some deliveries that might be amenable to forceps delivery can be conducted to reduce caesarean section rates. In our study, all the vaginal deliveries were conducted by midwives and none of the OVDs was performed by the midwife. This result highlights the need to train midwives in the use of OVDs especially vacuum delivery. Indeed vacuum delivery is a component of essential obstetric care and it is a very useful midwifery skills that can reduce maternal and perinatal morbidity and mortality [3].

Women who had OVDs were averagely younger than women who had vaginal deliveries. Furthermore, the likelihood of having OVDs generally decreases with age. This finding may be related to the fact that young parturient are likely to be nulliparous with uterine inertia and untested or rigid pelvis [15]. Moreover, our study also revealed that the odds of OVDs decreases with increasing parity. Thus, in order to reduce primary and subsequent repeat caesarean sections among young nulliparous women, the availability of a skilled accoucheur that can perform OVDs among well selected patients cannot be over-emphasized. Similarly younger women and nulliparous women had higher likelihood of having forceps delivery as compared to vacuum delivery, possibly because of exhaustion during prolonged second stage [15,41,42].

From our cross-sectional study, the prevalence of OVDs was higher among women that were underweight, while obese patients had the highest prevalence of vaginal deliveries. We also found that the odds of OVDs reduces with increasing BMI as was also reported by Yu and Wue in China and other authors [43,44]. This finding may also suggest that patients with higher BMI may have very roomy pelvis and may not require assisted delivery. This observed reduced odds of OVDs with increasing BMI will only be valid among women with adequate pelvis, as one of the requirements for OVD is a clinically adjudged adequate pelvis with no feature of cephalopelvic disproportion [45]. However, this pattern may not be absolute as increased birthweight and increased prevalence of medical conditions in pregnancy such as hypertension, cardiac disease and diabetes may be commoner among obese women thereby leading to increased caesarean section rates among obese women [43,46,47,48]. Since one of the indications for OVDs is prolonged second stage of labor, it may not be surprising that there were 11% higher odds of OVDs for every extra hour of labor. Therefore benefits of prevention of primary caesarean section by extending second stage of labor must be weighed against potential adverse maternal and neonatal outcomes [49].

## 5. Conclusions

This study has highlighted the downward trend of OVD over the past 15 years at a rate of 1.5% and few OVDs were performed by consultant and none done by the midwives which might contribute to the dying art of the obstetric emergency skill in our Centre’s. In order to revive the training opportunities on managing second stage labor which is incorporated in the guideline of Essential Steps in the Management of Obstetric Emergencies(ESMOE), Regular drills and use of mannequins, training of midwives on how to perform OVDs, increased training interest by consultants, and availability of equipment for OVDs can help revive the dying art of OVDs.

## Figures and Tables

**Figure 1 ijerph-19-16182-f001:**
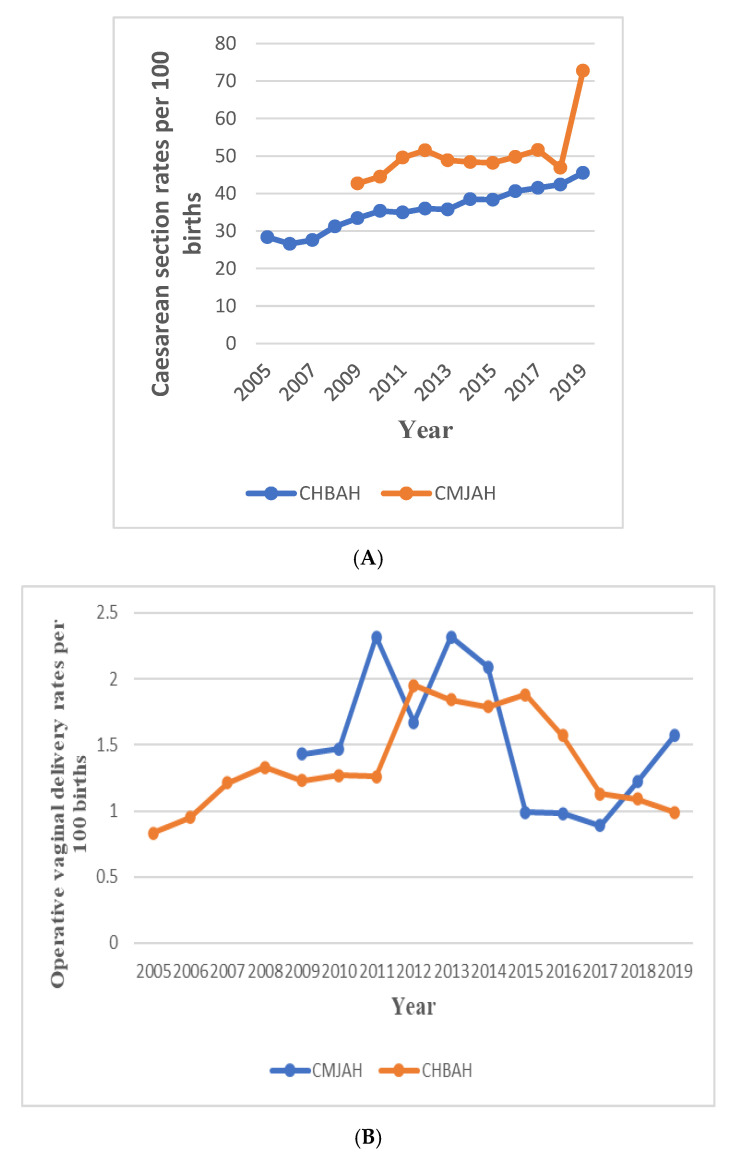
Trends in caesarean section (**A**) and operative vaginal delivery (**B**).

**Figure 2 ijerph-19-16182-f002:**
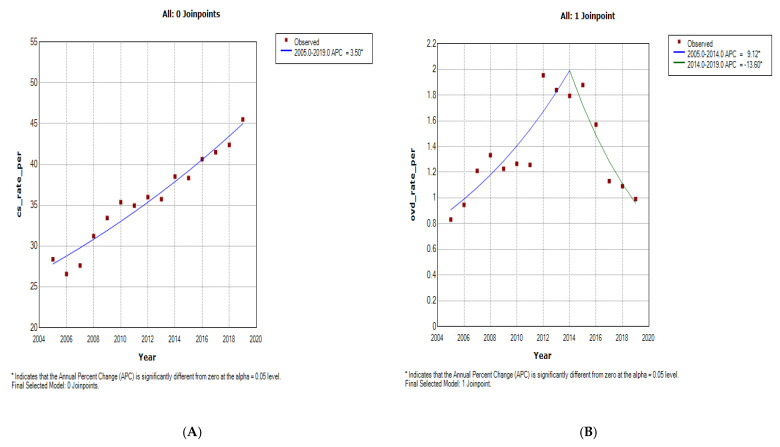
Join point regression analysis of the trends in caesarean section (**A**) and operative vaginal delivery rates (**B**) at Chris Hani Baragwanath Academic Hospital (2005–2019).

**Figure 3 ijerph-19-16182-f003:**
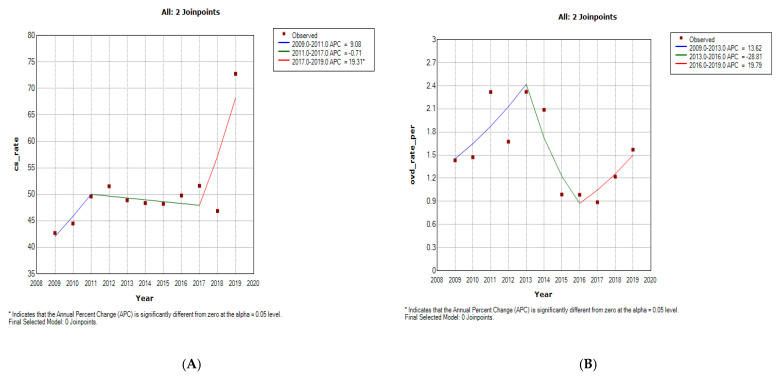
Join point regression analysis of the trends in caesarean section (**A**) and operative vaginal delivery (**B**) rates at Charlotte Maxeke Johannesburg Academic Hospital (2009–2019).

**Figure 4 ijerph-19-16182-f004:**
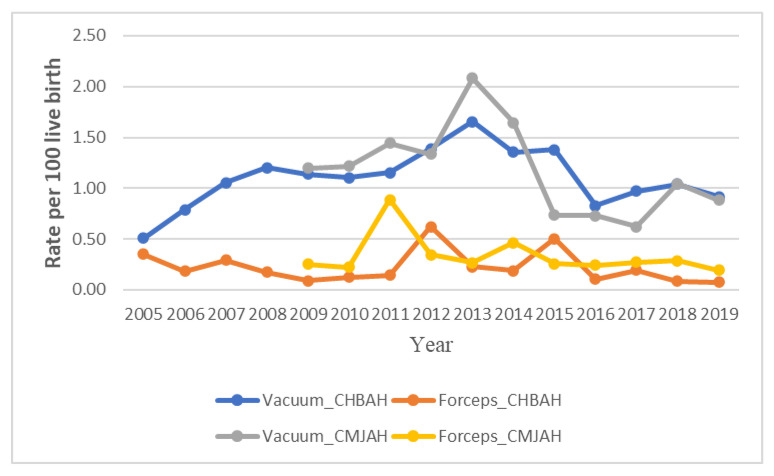
Trends in types of operative vaginal delivery rate at two academic hospitals, Johannesburg (2005–2019).

**Figure 5 ijerph-19-16182-f005:**
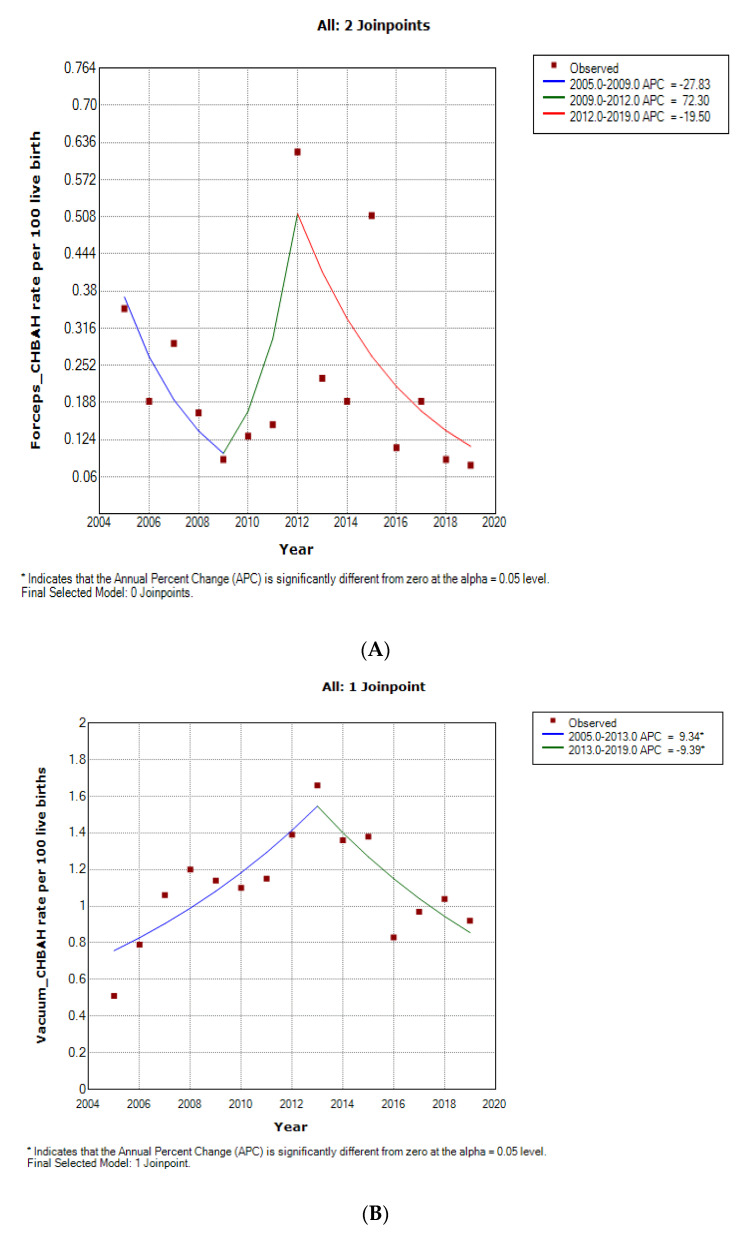
Join point regression analysis of the trends in forceps delivery (**A**) and Vacuum (**B**) rates at Chris Hani Baragwanath Academic Hospital (2005–2019).

**Figure 6 ijerph-19-16182-f006:**
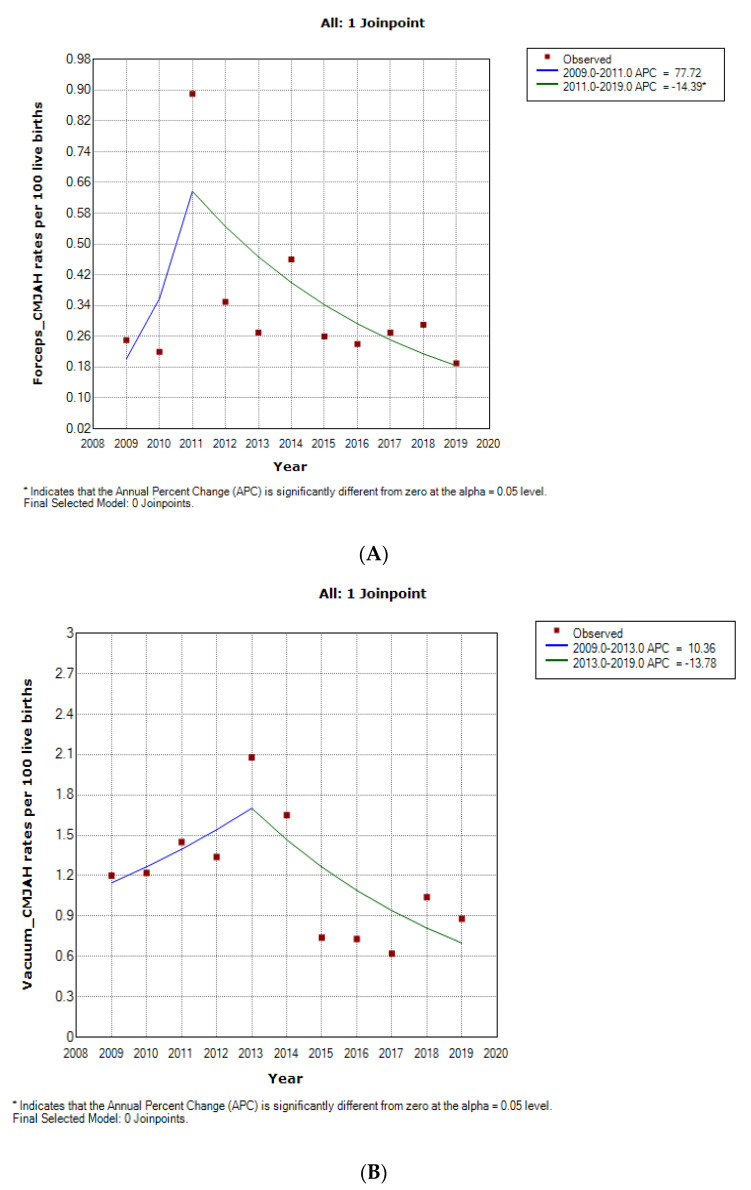
Join point regression analysis of the trends in Forceps deliveries (**A**) and Vacuum (**B**) rates at Charlotte Maxeke Johannesburg Academic Hospital, 2009–2019.

**Figure 7 ijerph-19-16182-f007:**
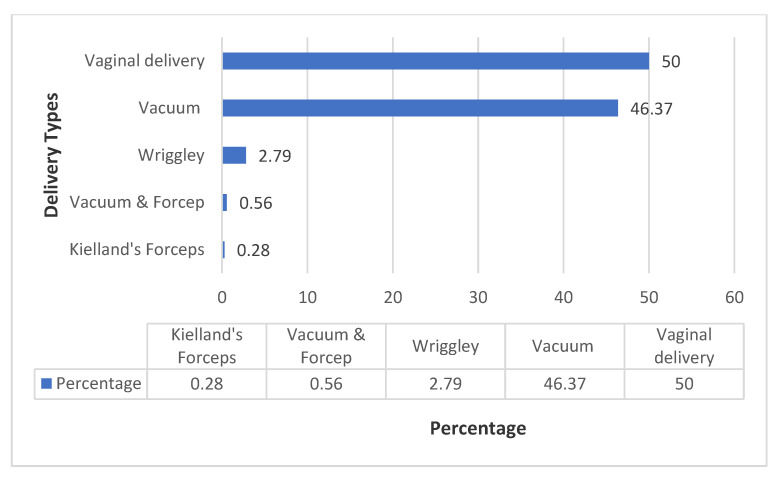
Types of operative vaginal delivery at the two hospitals.

**Table 1 ijerph-19-16182-t001:** Join point regression of the trends in the caesarean section and operative vaginal deliveries at two academic hospitals in Johannesburg (2005–2019).

Hospital and Type of Delivery	Year	APC (%) 95%CI	AAPC 95%CI	*p*-Value	Comment
Caesarean section					
CHBAH					
Overall	2005–2019	-	3.5 (3–4)	<0.001	Statistically significant increase
CMJAH					
Trends 1	2009–2011	9.1 (−4.5 to 24.6)		0.1	Non-Statistically significant increase
Trends 2	2011–2017	−0.7 (−3.4 to 2.1)		0.5	Non-Statistically significant decrease
Trends 3	2017–2019	19.3 * (6.2 to 34.1)		<0.001	Statistically significant increase
Overall	2009–2019		4.1 (−10.7 to 2.9)	0.2	Non-Statistically significant increase
Operative Vaginal delivery					
CHBAH					
Trends 1	2005–2014	9.1 * (5.4–13.0)		<0.001	Statistical significant increase
Trends 2	2014–2019	−13.6 *		<0.001	
Overall	2005–2019	-	−0.4 (−2.6 to 3.7)	0.8	Non-statistically significant decrease
CMJAH					
Trends 1	2009–2013	13.6 (54.0 to 1.3)		0.3	Non-statistically significant increase
Trends 2	2013–2016	−28.8 (−74.8 to 101.3)		0.4	Non-statistically significant decrease
Trends 3	2016–2019	19.8 (−31.4 to 109.1)		0.4	Non-statistical significant icrease
Vacuum delivery					
CHBAH					
Trends 1	2005–2013	9.3 (2.9 to 16.2)		<0.001	Statistically significant increase
Trends 2	2005–2019	−9.4 (−5.6 to −1.3)		<0.001	Statistically significant decrease
Overall	2005–2019		0.9 (−3.5 to 5.5)	0.7	Non-statistically significant increase
CMJAH					
Trends 1	2009–2013	10.4 (−17.2 to 2.9)		0.4	Non-statistically sigificant increase
Trends 2	2013–2019	−13.8(−26.8 to 1.5)		0.1	Non-statistically significant decrease
Overall	2009–2019		−4.8 (−15.7 to 7.4)	0.4	Non-statistically Significant decrease
Forceps Delivery					
CHBAH					
Trends 1	2005–2009	−27.8 (−58.3 to 47.1)		0.2	Non-statistically significant decrease
Trends 2	2009–2012	72.3 (−83.1 to 16.5)		0.6	Non-statistically significant increase
Trends 3	2012–2019	−19.5 (−40.9 to 9.6)		0.1	Nons-statistically significant decrease
	2005–2019		−8.2 (−41.5 to 44.1)	0.7	Non-statistically significant decrease
CMJAH					
Trends 1	2009–2011	77.7 (−30.0 to 35.13)		0.2	Non-statistically significant increase
Trends 2	2011–2019	−14.4 (−22.0 to −6.0)		<0.001	Statistically significant decrease
Overall	2009–2019		−0.9 (−15.7 to 16.4)	0.9	Non-statistically significant decrease

* Statistically significant level at *p*-value < 0.005. OVD: Operative vaginal delivery; C/S: Caesarean Section; CBAH: Chris Hani Baragwanath Academic Hospital; CMJAH: Charlotte Maxeke Johannesburg Academic Hospital.

**Table 2 ijerph-19-16182-t002:** Comparison of the socio-demographic characteristics of patients who had vaginal delivery and those who had operative vaginal delivery from the two academic hospitals. In Johannesburg.

Characteristics	Vaginal Delivery *n* = 179, (%)	OVD *n* = 179, (%)	Total *n* = 358, (%)	*p*-Value
Hospital				
CHBAH	92 (51.39)	92 (51.39)	184 (51.39)	1.000
CMJAH	87 (48.60)	87 (48.60)	174 (48.60)	
Age (Years) Mean, SD	28.68 ± 6.78	24.92 ± 5.74	26.80 ± 6.55	<0.0001
<20	15 (8.38)	26 (14.53)	41 (11.45)	<0.0001
20–24	35 (19.55)	79 (44.13)	114 (31.84)	
25–29	51 (28.49)	35 (19.55)	86 (24.02)	
30–34	38 (21.23)	25 (13.97)	63 (17.60)	
≥35	40 (22.35)	14 (7.82)	54 (15.08)	
Age (Years)				
<35	139 (77.65)	165 (92.18)	304 (84.92)	<0.0001
≥35	40 (22.35)	14 (7.82)	54 (15.08)	
Ethnic group				
Black	171 (95.50)	167 (93.29)	338 (94.41)	0.803
Coloured	4 (2.23)	5 (2.79)	9 (2.51)	
Indian	3 (1.67)	5 (2.79)	8 (2.23)	
White	1 (0.55)	2 (1.11)	3 (0.84)	
Ethnic group				
Blacks	171 (95.53)	167 (93.29)	338 (94.41)	0.357
Others	8 (4.47)	12 (6.70)	20 (11.17)	
Employment status				
Employed	42 (23.46)	27 (15.08)	69 (38.55)	0.046
Unemployed	118 (65.92)	121 (67.59)	239 (66.76)	
Student	19 (10.61)	31 (17.32)	50 (13.97)	
Marital status				
Married	17 (9.49)	17 (9.49)	34 (9.49)	1.000
Single	162 (90.50)	162 (90.50)	324 (90.50	
Gestational age	39 (38–40)	39 (38–40)	39 (38–40)	0.8689
27	0 (0.00)	2 (1.12)	2 (0.56)	0.756
34	1 (0.56)	0 (0.00)	1 (0.28)	
35	2 (1.12)	2 (1.12)	4 (1.12)	
36	5 (2.79)	3 (1.68)	8 (2.23)	
37	13 (7.26)	15 (8.38)	28 (7.82)	
38	39 (21.79)	35 (19.55)	74 (20.67)	
39	48 (26.82)	49 (27.37)	97 (27.09)	
40	39 (21.79)	44 (24.58)	83 (23.18)	
41	28 (15.64)	21 (11.73)	49 (13.69)	
42	4 (2.23)	8 (4.47)	12 (3.35)	
Gestational age (weeks)				
<37	8 (4.47)	7 (3.91)	15 (4.19)	0.792
≥37	171 (95.53)	172 (96.09)	343 (95.81)	
Parity (median, IQR)	1 (0–2)	0 (0–1)	1 (0–2)	<0.001
0	51 (28.49)	108 (60.33)	159 (44.41)	<0.001
1	48 (26.81)	57 (31.84)	105 (29.33)	
2	41 (22.90)	6 (3.35)	47 (13.13)	
3	27 (15.08)	6 (3.35)	33 (9.22)	
4	5 (2.79)	2 (0.11)	7 (1.95)	
5	5 (2.79)	0 (0)	5 (1.39)	
6	2 (0.11)	0 (0)	2 (0.55)	
Parity Category				
0	51 (28.49)	108 (60.34)	159 (44.41)	<0.001
1–3	116 (64.80)	69 (38.55)	185 (51.68)	
≥4	12 (6.70)	2 (1.12)	14 (3.91)	
Body Mass Index (kg/m^2^)				
Underweight (<18.5)	1 (0.55)	5 (2.79)	6 (1.68)	0.001
Normal (18.5–24.9)	63 (35.19)	86 (48.04)	149 (41.62)	
Overweight (25–29.9)	45 (25.14)	51 (28.49)	96 (26.82)	
Obese (≥30)	70 (39.10)	37 (20.67)	107 (29.89)	
Booking status				
Unbooked	6 (3.35)	4 (2.23)	10 (2.79)	0.521
Booked	173 (96.65)	175 (97.77)	348 (97.20)	
Number of antenatal visits Median, IQR	5 (4–6)	5 (4–7)	5 (4–7)	0.2746
HIV status				
Negative	137 (76.54)	154 (86.03)	291 (81.28)	0.021
Positive	42 (23.46%)	25 (13.97%)	67 (18.72)	
HIV positive participants only				
CD4 count median, IQR (cells/mL)	413 (325–527)	329.5 (250–447)	387 (292–506)	0.0598
<250	6 (13.95)	6 (23.08)	12 (17.39)	0.122
250–349	8 (18.60)	10 (38.46)	18 (26.09)	
350–499	15 (34.88)	6 (23.08)	21 (30.43)	
≥500	14 (32.56)	4 (15.38)	18 (26.09)	
Medical morbidity				
Yes	29 (16.20)	30 (16.76)	59 (16.48)	0.887
No	150 (83.80)	149 (83.24)	299 (83.52)	
Induction of labor				
yes	32	20	52 (14.53)	0.072
no	147	159	306 (85.47)	
Duration of labor				
Median, IQR	7 (5–9)	9 (6–11)	8 (6–10)	<0.001
Cadre of the accoucheur				
Midwife	179 (100.00)	0 (0.00)	179 (50.00)	<0.001
Medical officer	0 (0.00)	16 (8.94)	16 (4.47)	
Registrar	0 (0.00)	154 (86.03)	154 (43.02)	
Consultant	0 (0.00)	9 (5.03)	9 (2.51)	

**Table 3 ijerph-19-16182-t003:** Univariable and multivariable regression of the association between sociodemographic and clinical characteristics and operative vaginal delivery.

Factor	Univariable			Multivariable		
	COR	95%CI	*p*-Value	Ad OR	95%CI	*p*-Value
Hospital						
CHBAH	1.00	Reference	Reference			
CMJAH	1.00	0.66–1.51	1.00	1.10	0.69–1.74	0.694
Age (Years)						
<20	1.00	Reference	Reference	1.00	Reference	Reference
20–24	1.30	0.61–2.76	0.491	1.95	0.89–4.28	0.097
25–29	0.40	0.18–0.85	0.018	0.77	0.33–1.79	0.548
30–34	0.38	0.17–0.86	0.019	0.97	0.36–2.60	0.953
≥35	0.20	0.08–0.49	<0.001	0.61	0.22–1.72	0.354
Age (Years)						
<35	1.00	Reference	Reference	-	-	-
≥35	0.29	0.15–0.56	<0.001	-	-	---
Gestational age (weeks)						
<37	1.00	Reference	Reference			
≥37	1.15	0.41–3.24	0.792			
Parity	0.43	0.33–0.56	<0.001			
Parity Category						
0	1.00	Reference	Reference	1.00	Reference	Reference
1–3	0.28	0.18–0.44	<0.001	0.42	0.24–0.74	0.002
≥4	0.08	0.02–0.37	0.001	0.13	0.02–0.81	0.029
Body Mass Index	0.94	0.91–0.98	0.002			
Underweight (<18.5)	1.00	Reference	Reference	1.00	Reference	Reference
Normal (18.5–24.9)	0.27	0.03–2.40	0.242	0.19	0.04–0.91	0.037
Overweight (25–29.9)	0.23	0.03–2.02	0.183	0.19	0.04–0.96	0.045
Obese (≥30)	0.11	0.01–0.94	0.044	0.11	0.02–0.54	0.007
Ethnic group						
Black	1.00	Reference	Reference	-	-	-
Coloured	1.28	0.34–4.86	0.717	-	-	---
Indian/Asian	1.71	0.40–7.27	0.470	-	-	-
White	2.05	0.18–22.88	0.560	-	-	---
Ethnic groups						
Others	1.00	Reference	Reference	-	-	-
Blacks	0.65	0.26–1.64	0.361	-	-	---
Employment status						
Employed	1.00	Reference	Reference	-	-	-
Unemployed	1.60	0.92–2.76	0.094	-	-	---
Student	2.54	1.20–5.37	0.015			
Marital status						
Married	1.00	Reference	Reference	-	-	-
Single	1.00	0.49–2.03	1.000	-	-	---
Booking status						
Unbooked	1.00	Reference	Reference			
Booked	1.52	0.42–5.48	0.525	-	-	-
HV status				-	-	---
Positive	1.00	Reference	Reference			
Negative	1.89	1.85–1.93	<0.001			
Medical co-morbidity						
Yes	1.00	Reference	Reference			
No	0.96	0.89–1.04	0.321			
Induction of labor						
Yes	1.00	Reference	Reference			
No	0.58	0.32–1.06	0.075			
Duration of labor	1.17	1.10–1.25	<0.001	1.11 ^	0.004	1.03–1.20

COR: Crude odds ratio; ad OR: Adjusted odds ratio. ^: odds ratio obtained from model II. CBAH: Chris Hani Baragwanath Academic Hospital; CMJAH: Charlotte Maxeke Johannesburg Academic Hospital.

**Table 4 ijerph-19-16182-t004:** Comparison of the sociodemographic and clinical characteristics among women that had forceps and vacuum deliveries.

Characteristics	Forceps *n* = 11 (%)	Vacuum *n* = 166 (%)	Total *n* = 177 (%)	*p*-Value
Hospital				
CHBAH	9 (81.82)	83 (50.00)	92 (51.98)	0.060
CMJAH	2 (18.18)	83 (50.00)	85 (48.02)	
Age mean ± SD (Years)	23.91 ± 6.46	24.99 ± 5.73	24.93 ± 5.77	0.547
<20	3 (27.27)	23 (13.86)	26 (14.69)	0.252
20–24	4 (36.36)	74 (44.58)	78 (44.07)	
25–29	2 (18.18)	32 (19.28)	34 (19.21)	
30–34	0 (0.00)	25 (15.06)	25 (14.12)	
≥35	2 (18.18)	12 (7.23)	14 (7.91)	
Age (Years)				
<35	9 (81.82)	154 (92.77)	163 (92.09)	0.212
≥35	2 (18.18)	12 (7.23)	14 (7.91)	
Ethnic group				
Black	11 (0.00)	154 (3.01)	165 (2.82)	1.000
Coloured	0 (0.00)	5 (3.01)	5 (2.82)	
Indian	0 (0.00)	5 (1.20)	5 (1.13)	
White	0 (100.00)	2 (92.77)	2 (93.22)	
Ethnic group				
Blacks	11 (100.00)	154 (92.77)	165 (93.22)	1.000
Others	0 (0.00)	12 (7.23)	12 (6.78)	
Employment status				
Employed	1 (9.09)	26 (15.66)	27 (15.25)	0.055
Unemployed	5 (45.45)	114 (68.67)	119 (67.23)	
Student	5 (45.45)	26 (15.66)	31 (17.51)	
Marital status				
Married	0 (0.00)	17 (10.24)	17 (9.60)	0.603
Single	11 (100.00)	149 (89.76)	160 (90.40)	
Gestational age (median, IQR) weeks	39 (37–40)	39 (38–40)	39 (38–40)	0.2905
27	2 (18.18)	0 (0.00)	2 (1.13)	0.027
35	0 (0.00)	2 (1.20)	2 (1.13)	
36	0 (0.00)	3 (1.81)	3 (1.69)	
37	1 (9.09)	14 (8.43)	15 (8.47)	
38	1 (9.09)	33 (19.88)	34 (19.21)	
39	4 (36.36)	45 (27.11)	49 (27.68)	
40	2 (18.18)	41 (24.70)	43 (24.29)	
41	0 (0.00)	21 (12.65)	21 (11.86)	
42	1 (9.09)	7 (4.22)	8 (4.52)	
Gestational age				
<37	2 (18.18)	5 (3.01)	7 (3.95)	0.062 0.012
≥37	9 (81.82)	161(96.99)	170 (96.05)	
Parity	0 (0–0)	0 (0–1)		0.111
0	9 (81.82)	97 (58.43)	106 (59.89)	0.737
1	2(18.18)	55(33.13)	57 (32.20)	
2	0 (0.00)	6 (3.61)	6 (3.39)	
3	0 (0.00)	6 (3.61)	6 (3.39)	
4	0 (0.00)	2 (1.20)	2 (1.13)	
5	0 (0.00)	0 (0.00)	0 (0.00)	
6	0 (0.00)	0 (0.00)	0 (0.00)	
Parity Category				
0	9 (81.82)	97 (58.43)	106 (59.89)	0.305
1–3	2 (18.18)	67 (40.36)	69 (38.98)	
≥4	0 (0.00)	2 (1.20)	2 (1.13)	
Body Mass Index	23.1 (21.8–28.5)	24.95 (22.7–28.9)	26.1 (23.1–31.2)	0.716
Underweight (<18.5)	0 (0.00)	5 (3.01)	5 (2.82)	1.000
Normal (18.5–24.9)	6 (54.55)	78 (46.99)	84 (47.46)	
Overweight (25–29.9)	3 (27.27)	48 (28.92)	51 (28.81)	
Obese (≥30)	2 (18.18)	35 (21.08)	37 (20.90)	
Booking status				
Unbooked	0 (0.00)	4 (2.41)	4 (2.26)	1.000
Booked	11 (100.00)	162 (97.59)	173 (97.74)	
HV status				
Negative	10 (90.91)	142 (85.54)	152 (85.88)	1.000
Positive	1 (9.09)	24 (14.46)	25 (14.12)	
Accoucheur				
Medical officer	0 (0.00)	16 (9.64)	16 (9.04)	0.020
Registrar	8 (72.73)	144 (86.75)	152 (85.88)	
Consultant	3 (27.27)	6 (3.61)	9 (5.08)	
Indication				
Maternal	7 (63.64)	94 (56.63)	101 (57.06)	0.677
Fetal	2 (18.18)	50 (30.12)	52 (29.38)	
Both	2 (18.18)	22 (13.25)	24 (13.56)	
Maternal Indication				
Delayed second stage	5 (62.50)	63 (54.31)	68 (54.84)	0.198
Poor maternal effort	0 (0.00)	30 (25.86)	30 (24.19)	
Delayed 2nd stage and poor maternal effort	3 (37.50)	23 (19.83)	26 (20.97)	
Fetal indication				
Fetal distress	3 (75.00)	73 (100.00)	76 (98.70)	0.052
Preterm/prematurity	1 (25.00)	0 (0.00)	1 (1.30)	
Induction of labor				
Yes	1 (9.09)	19 (11.45)	20 (11.30)	1.000
No	10 (90.91)	147 (88.55)	157 (88.70)	
Duration of labor (median, IQR) hours	10 (6–11)	9 (6–11)	9 (6–11)	0.913
Position of the presenting part				
ROA	3 (27.27)	63 (37.95)	66 (37.29)	0.331
ROP	1 (9.09)	2 (1.20)	3 (1.69)	
ROT	7 (63.64)	91 (54.82)	98 (55.37)	
LOA	0 (0.00)	9 (5.42)	9 (5.08)	
LOP	0 (0.00)	1 (0.60)	1 (0.56)	
Analgesia				
Epidural	0 (0.00)	1 (0.60)	1 (0.56)	
Local anesthetic	5 (45.45)	75 (45.18)	80 (45.20)	0.441
Pethidine only	1 (9.09)	5 (3.01)	6 (3.39)	
Local anesthetics and Pethidine	5 (45.45)	85 (51.20)	90 (50.85)	

CBAH: Chris Hani Baragwanath Academic Hospital; CMJAH: Charlotte Maxeke Johannesburg Academic Hospital. IQR: Interquartile range; SD: Standard deviation.

**Table 5 ijerph-19-16182-t005:** Univariable and multivariable regression of the association between sociodemographic and clinical characteristics and vacuum delivery among women who had operative vaginal delivery.

Factor	Univariable			Multivariable		
	OR	95%CI	*p*-Value	Ad OR	95%CI	*p*-Value
Hospital						
CHBAH	1.00	Reference	Reference	1.00	Reference	Reference
CMJAH	4.5	0.94–21.46	0.059	2.90	0.61–13.77	0.18
Age (Years)						
<20	1.00	Reference	Reference	1.00	Reference	Reference
20–24	2.41	0.50–11.64	0.273			
25–29	2.09	0.32–13.59	0.442			
30–34	1.00	-	-			
≥35	0.78	0.11–5.37	0.803			
Age (Years)						
<35	1.00	Reference	Reference	1.00	Reference	Reference
≥35	0.35	0.07–1.82	0.212	0.14	0.02–0.81	0.029
Gestational age						
≥37	1.00	Reference	Reference	1.00	Reference	Reference
<37	0.14	0.02–0.82	0.030	0.36	0.09–1.43	0.146
Parity	2.91	0.78–10.83	0.112			
Parity Category						
0	1.00	Reference	Reference	1.00	Reference	Reference
≥1	3.20	0.67–15.35	0.146	7.51	1.99–28.32	0.003
Accoucheur						
Consultant	1.00	Reference	Reference	1.00	Reference	Reference
Registrar/Medical officer	10	2.10–47.67	0.004	7.11	1.29–39.02	0.024
Body Mass Index						
Underweight (<18.5)						
Normal (18.5–24.9)	1.00	Reference	Reference			
Overweight (25–29.9)	1.23	0.29–5.17	0.777			
Obese (≥30)	1.35	0.26–7.04	0.725			
HV status						
Negative	1.00	Reference	Reference			
Positive	1.69	0.21–13.89	0.625			
Indication						
Both	1.00	Reference	Reference			
Maternal	1.22	0.24–6.28	0.811			
Fetal	2.27	0.30–17.19	0.426			
Employment status						
Employed	1.00	Reference	Reference			
Unemployed	0.88	0.10–7.88	0.907			
Student	0.20	0.02–1.84	0.156			

CBAH: Chris Hani Baragwanath Academic Hospital; CMJAH: Charlotte Maxeke Johannesburg Academic Hospital. IQR: Interquartile range; SD: Standard deviation; COR: Crude odds ratio; ad OR: Adjusted odds ratio.

## Data Availability

The data presented in this study are available on request from the corresponding author. The data are not publicly available due to ethical restrictions.

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
