# Peer review of "Trends and Determinants of Operative Vaginal Delivery at Two Academic Hospitals in Johannesburg, South Africa 2005–2019"

_ijerph, 2022, doi:10.3390/ijerph192316182_

Round 1
Reviewer 1 Report
The manuscript is interesting. The information is well documented and presented and reflects a trend of decreasing OVD’s and incresing C-sections.
I only have one aspect that needs clarification: Lines 238-240 All the vaginal delivery were conducted by the mid- 238 wives, but no midwife conducted any OVD. Of the 179 OVDs, majority (n=154, 86.03%) 239 were performed by registrars.
Are midwifes allowed to perform OVD in South Africa? In many countries OVD is beyond the competence of midwifes.
Author Response
Humble greeting to Olivia and editorial team, apologies for late uploading of the corrected version of the manuscript. I must take the opportunity we have reached with journal and thanks so much for the humble remarks to improve the manuscript. I have combined the comments in one word document for reviewer 1 and 2, apologies if this may result in any inconvenience from their reviewers. I had also attached highlighted manuscript and clean manuscript.
Regards
Dr Dutywa Afikile

Reviewer 2 Report
Dear Author,
I have read with great interest your paper on OVD trends.
Please find below a few comments:
1. Results section 3.2- needs shortening; the same data on cs trends are shown on a graph, and table and then discussed in the main text. Please do not repeat the same results in tables and in the text.
2. The same remark applies to the sections on OVD trends. To sum up I understand there was no significant change in the modes of delivery in the studied period?
3. Section 3.6 - first sentence repeats the methods section on the review of medical records; I feel this repetition is unnecessary
4. section 3.7 again describes the characteristic of the total population, which is already in the table. The repetition is unnecessary. Maybe just a general summary of the characteristics is enough. The details could be in the comparison between the study and control group, which I find in section 3.8
5. Section 3.8 again repeats the details of the table, please consider making it more comprehensive
6. section 3.9 needs a clear statement what were the risk factors for OVD in your study, the table might be is supplementary material, the section is unclear and needs rewriting
7. Could you elaborate on prematurity as a fetal indication for OVD?
8. discussion lines 359-366 there is a lacking fragment of sentences?
9. discussion line 401 suggests that OVD is a good choice for an inadequate pelvis, which explains higher rates of OVD in slim women? as far as my experience is concerned the OVD is contraindicated in case of inadequate pelvis
Author Response
Thank you so much for the humble comments, i have attached the response that have helped us a lot to improve the manuscript. Hope this will find you well.
Regards
Dr Afikile Dutywa

Round 2
Reviewer 2 Report
Dear Authors,
Thank you for your responses to my comments. I recommend publishing your paper after minor editing (mainly spell-check).
Best regards,